# A Comparison of Functional and Oncologic Outcomes between Partial Nephrectomy and Radiofrequency Ablation in Patients with Chronic Kidney Disease after Propensity Score Matching

**DOI:** 10.3390/diagnostics12102292

**Published:** 2022-09-23

**Authors:** Hyunsoo Ryoo, Minyong Kang, Hyun Hwan Sung, Hwang Gyun Jeon, Byong Chang Jeong, Seong Soo Jeon, Hyun Moo Lee, Byung Kwan Park, Seong Il Seo

**Affiliations:** 1Department of Urology, VHS Medical Center, Seoul 05368, Korea; 2Department of Urology, Samsung Medical Center, Sungkyunkwan University School of Medicine, 81 Irwon-ro, Gangnam-gu, Seoul 06351, Korea; 3Department of Radiology, Samsung Medical Center, Sungkyunkwan University School of Medicine, Seoul 06351, Korea

**Keywords:** chronic kidney disease, partial nephrectomy, radiofrequency ablation, propensity score matched analysis, functional and oncological outcomes

## Abstract

Purpose: This study aimed to compare functional and oncological outcomes between partial nephrectomy (PN) and radiofrequency ablation (RFA) for a small renal mass (SRM, ≤4 cm) in patients with chronic kidney disease (CKD). Materials and Methods: Patients with CKD who underwent either PN or RFA for SRM between 2005 and 2019 were included. Patients were stratified into two categories: CKD stage 2 and CKD stage 3 or higher. We performed propensity score matching (PSM) analysis in patients with CKD stage 2 and CKD stage 3 or higher. We compared the functional and oncological outcomes between two groups according to CKD stage before and after PSM. Results: Among 1332 patients, 1195 patients were CKD stage 2 and 137 patients were CKD stage 3 or higher. After PSM analysis using age, pre-treatment eGFR, and clinical tumor size as matching variables, the PN and RFA groups had 270 and 135 CKD stage 2 patients, respectively, and both had 53 patients each with CKD stage 3 or higher. There were no significant differences in percent change in eGFR at 1 year post-operation between groups in patients with CKD stage 2 and stage 3 or higher. Among all patients with tissue-proven malignancy, the 5-year recurrence-free survival (RFS), cancer-specific survival, and overall survival were significantly higher in the PN group. However, only the 5-year RFS was significantly higher in the PN group after matching. Conclusion: Mortality is low in patients with SRM, and functional outcomes were not significantly different between the two treatments. RFA could be an alternative treatment modality in patients who are poor candidates for surgery.

## 1. Introduction

Due to the rapid development and widespread use of modern imaging modalities, the detection of small renal masses continues to increase [1,2,3]. For patients with a small renal mass, especially those in clinical stage T1a, multiple guidelines have emphasized the use of nephron-sparing treatment [4,5,6,7]. Partial nephrectomy (PN) remains the gold standard treatment modality for patients with small renal tumors [8]. However, PN has some limitations in patients who are poor candidates for surgery as it is associated with potential perioperative complications [3,9,10,11]. Ablative procedures are recommended in the management of small renal masses in patients with significant comorbidities who are poor surgical candidates [4,8,12]. Radiofrequency ablation (RFA) is increasingly used in the treatment of small renal tumors, especially in patients with comorbidities [3,8]. Of the various comorbidities, chronic kidney disease (CKD) demands the most attention, as patients with CKD are vulnerable to deterioration in renal function after treatment. Although there are several studies comparing functional and oncological outcomes between PN and RFA in general patients, there are few studies that report those outcomes in CKD patients [4,8,13,14,15]. Therefore, in the present study, we aimed to perform propensity score matched comparative analyses of the functional and oncological outcomes for PN vs. RFA in CKD patients that could aid clinicians and patients to identify the optimal treatment strategies.

## 2. Materials and Methods

### 2.1. Patients

After Institutional Review Board approval, we collected clinical data of CKD patients who underwent PN or percutaneous RFA for a small renal mass at our center between 2005 and 2019. CKD was defined as an estimated glomerular filtration rate (eGFR) of less than 90 mL/min/1.73 m^2^ and was estimated from calibrated serum creatinine measurements using the Chronic Kidney Disease Epidemiology Collaboration equation. CKD stages > 1 were considered for analysis. Patients presenting with multiple renal masses, bilateral renal masses, or with a history of hereditary renal cell carcinoma (RCC) were excluded from the analysis. Furthermore, patients who underwent radical nephrectomy, partial nephrectomy, or ablation therapy on their kidneys previously were excluded. A total of 1332 patients treated with PN (*n* = 1144) and RFA (*n* = 188) for cT1aN0M0 renal masses were included.

### 2.2. Patient Management

According to the selection criteria for PN and RFA, a renal mass was assessed by a urologist to determine if partial resection was possible. Patients who refused the recommended PN, had tumors in unresectable locations, or who were unable to undergo surgery because of other underlying diseases were recommended treatment with RFA. Open or laparoscopic PN (laparoscopic PN, hand-assisted laparoscopic PN, and robot-assisted laparoscopic PN) was performed by seven experienced urologists. The surgical techniques have been previously described [16,17,18]. After clamping of the renal artery during PN, the tumor was excised with a cold scissor outside the zone of a 0.5-centimeter peritumoral margin. The tumor bed was closed using 3–0 Vicryl sutures and then renorrhaphy was performed using 2–0 Vicryl sutures.

RFA was performed by 2 experienced uroradiologists using computed tomography (CT) guidance (Aquilion, Toshiba Medical Systems, Otawara, Japan). A Cool-tip RF electrode (Radionics, Inc., Burlington, MA, USA) or Proteus RF electrode (STARmed, Goyang, Gyeonggi-do, Korea) was used for RFA. Ablations were overlapped to cover a tumor margin (more than 5 mm) as well as a tumor after the first control. The ablation area was monitored with repeated CT scans. Additional ablations were performed at the radiologists’ discretion if the previous ablation was considered incomplete in post-treatment CT imaging.

### 2.3. Clinical Features and Follow-Up

Clinical features included age, sex, mass size, eGFR, and pathological outcomes and complications. Histological diagnoses of PN and RFA were made using surgical specimens and biopsy cores, respectively.

Renal function data, estimated using the GFR from serum creatinine levels [14], were collected at the time of pre-treatment and at 1 year post-treatment. Functional outcome was evaluated by comparing these eGFR values.

### 2.4. Statistical Analysis

Demographic and clinical characteristics were analyzed using Student’s *t*-tests for continuous variables and the chi-squared test for categorical variables. Propensity scores were estimated using a logistic regression model, and propensity score matching analysis between the two groups was executed using R 3.6.1 (Vienna, Austria; http://www.R-project.org/) (accessed on 5 July 2019). A survival curve was generated using the Kaplan–Meier method, and differences between the PN and RFA groups were assessed using the log-rank test. The duration of follow-up for recurrence-free survival (RFS) was calculated from the treatment to recurrence or last follow-up. The overall survival (OS) and cancer-specific survival (CSS) were respectively defined as the proportion of patients who did not die from any cause and the proportion of patients who did not die from any cancer, including RCC. All statistical analyses were performed using SPSS version 21.0 (IBM Corp., Armonk, NY, USA), with a *p*-value < 0.05 considered statistically significant.

## 3. Results

### 3.1. Demographic and Clinical Characteristics

The clinicopathological features of the patients are described in Table 1. Among the CKD stage 2 patients (*n* = 1195), those who underwent PN were significantly younger than those treated with RFA (56 vs. 63 years; *p* < 0.001). The mean pre-treatment tumor size (2.37 vs. 2.13 cm; *p* < 0.001) as well as the mean pre-treatment eGFR (78.6 vs. 75.9 mL/min/1.73 m^2^; *p* < 0.001) were significantly higher in the PN group than in the RFA group. In patients with CKD stage higher than stage 2 (*n* = 137), the mean pre-treatment tumor size (2.52 vs. 2.20 cm; *p* = 0.022) and the mean pre-treatment eGFR (51.3 vs. 33.2 mL/min/1.73 m^2^; *p* < 0.001) were significantly higher in the PN group than in the RFA group.

### 3.2. Propensity Score Matching Analysis

For the propensity score matching analysis, we selected age, pre-treatment eGFR, and clinical tumor size as matching variables, which had shown significant differences between the PN and RFA groups. After matching, the number of CKD stage 2 patients in the PN and RFA groups was 270 and 135, respectively. There were no statistical differences in variables including age, pre-treatment eGFR, and clinical tumor size between the two treatment groups (Table 2). Furthermore, there were 53 patients each in the PN and RFA groups with CKD stage 3 or higher. There were no differences in variables between the two groups except in pre-treatment eGFR (48.6 vs. 33.2; *p* < 0.001).

### 3.3. Comparing Renal Functional Outcomes between Two Groups According to CKD Stage

Table 3 shows the comparative functional outcomes for the PN group vs. the RFA group in patients with CKD stage 2. In all patients with CKD stage 2, there were significant differences in eGFR pre-treatment and at 1 year post-treatment between the PN and RFA groups. In addition, there was a significant difference in the progression of CKD stage at 1 year post-treatment between the PN and RFA groups. However, in propensity score matched cohorts with CKD stage 2, there were no significant differences in eGFR pre-treatment and at 1 year post-treatment between the two groups. Furthermore, the mean percentage change in eGFR at 1 year postoperatively ((preoperative eGFR-postoperative 1-year eGFR)/preoperative eGFR x 100) was not significantly different between the PN and RFA groups (−6.1 vs. −5.5; *p* = 0.713).

Table 4 shows the comparative functional outcomes for the PN group vs. the RFA group in patients with CKD stage 3 or higher. There were significant differences in eGFR pre-treatment and at 1 year, 2 years, and 3 years post-treatment between the PN and RFA groups. However, the mean percentage changes in eGFR at 1 year and 2 years postoperatively were not significantly different between the PN and RFA groups. The mean percentage change in eGFR at 3 years postoperatively was significantly different between the two groups (−7.3 vs. −22.0; *p* = 0.012).

### 3.4. Comparing Oncological Outcomes between the PN and RFA Groups

Figure 1 shows the comparative oncological outcomes for the PN group vs. the RFA group in all patients with cT1a tissue-proven malignancy. The PN group showed significantly higher RFS, CSS, and OS than the RFA group (*p* < 0.001, *p* < 0.001, and *p* < 0.001, respectively).

Figure 2 shows the comparative oncological outcomes between the two treatment groups in the propensity score matched cohorts. There was a significant difference in RFS between the two groups (*p* < 0.001). However, no significant differences were observed in CSS and OS between the two groups (*p* = 0.586 and *p* = 0.054, respectively).

Figure 3 shows the comparative oncological outcomes between the two groups in the propensity score matched cohorts with CKD stage 2. The PN group showed significantly higher RFS than the RFA group (*p* = 0.001). However, no significant difference was observed in OS between the two groups (*p* = 0.052). There was no case of cancer-specific death in this cohort.

Figure 4 shows the comparative oncological outcomes between the two groups in the propensity score matched cohorts with CKD stage 3 or higher. The PN group showed significantly higher RFS than the RFA group (*p* = 0.001). However, no significant differences were observed in CSS and OS between the two groups (*p* = 0.513 and *p* = 0.799, respectively).

## 4. Discussion

In this study, there were no significant differences in the change in post-treatment renal function in patients with CKD stage 2 and stage 3 or higher between groups. Among all patients with tissue-proven malignancy, the 5-year RFS, CSS, and OS were significantly higher in the PN group. However, only the 5-year RFS was significantly higher in the PN group after propensity score matching. Compared with PN, RFA allows easier recovery, less surgical trauma, less estimated blood loss, and a shorter length of hospitalization, thereby reducing the recovery time and costs [19]. Therefore, RFA has been recognized as an acceptable treatment alternative in patients with an increased risk of surgical morbidity [20,21]. Comorbidities, particularly CKD, must be treated carefully, as the treatment may further deteriorate the kidney function. Therefore, herein, we investigated the functional and oncological outcomes for PN and RFA to evaluate their efficacy in treating small renal masses in patients with CKD.

PN and RFA are unavoidably followed by deterioration in renal functions. Thus, an additional decline in renal function is an important concern in patients with CKD receiving treatment for small renal tumors. Several previous studies reported the superiority of RFA to PN in preserving renal function after treatment [22,23,24]. However, our study showed that the decline in renal function was greater after RFA in all CKD patients. In line with this result, our previous study on the comparison between robotic PN and RFA for the treatment of cT1a RCC also showed a greater deterioration of renal functions after RFA [25]. In RFA, a change in renal function is closely related to the location of the RCC [25]. Endophytic RCC is likely to require a larger volume of tumor margins for ablation than exophytic RCC [26]. At our center, RFA was preferred for the treatment of endophytic renal mass over PN; thus, the RFA group might have a higher proportion of endophytic RCC than the PN group. Therefore, patients who underwent RFA may have had a greater decline in renal function because a greater volume was ablated during RFA. Although this study showed that the decline in renal function at 1 year post-treatment was greater after RFA in all CKD stage 2 patients, it was statistically insignificant (−4.3 vs. −5.5; *p =* 0.458). Additionally, in the propensity score matched cohorts with CKD stage 2, the differences in the mean percentage change in eGFR and CKD upstaging at 1 year post-treatment between the PN and RFA groups were not statistically significant. In patients with CKD stage 3 or higher, the mean percentage change in eGFR between the two groups was statistically significant at 3 years post-treatment (−7.3 vs. −22.0; *p* = 0.012), but not at 1 and 2 years post-treatment. Generally, it is likely that patients with poor general conditions who were vulnerable to renal functional deterioration were included in the RFA group. This result, showing a significant difference in the mean percentage change in eGFR between the two groups at 3 years post-treatment, seems to be due not to the effect of treatment modality but to the difference in characteristics between the two groups. Thus, we cannot conclude that RFA is less favorable for renal function preservation.

Regarding RFS, PN led to significantly higher RFS in patients regardless of propensity score matching and CKD stage. Recently, a study by Andrews et al. showed that the local recurrence and metastases were not statistically different among PN, RFA, and cryoablation for cT1a patients [4]. These findings are in agreement with the study by Olweny et al., who reported that 5-year local RFS and metastasis-free survival were statistically similar for RFA and PN in patients with cT1a RCC [27]. However, a systematic review and meta-analysis on the management of cT1 renal masses showed inferior local oncological control in patients treated with thermal ablation compared to patients treated with PN [28]. The reason for the different results among studies may be attributed to the variations in study design and patient groups. In our study, PN was associated with a higher RFS compared to RFA, which may be due to the following reasons. First, at our center, RFA is preferred for treating renal masses located near the renal hilum, which is difficult to treat with PN. However, a renal tumor near the renal hilum is difficult to ablate completely because of the renal sinus vessels and the fact that the collecting system is near the treating zone. Moreover, in the case of PN, the resected tumor margin can be checked, which is not possible with RFA. One study has reported an increase in the risk of local recurrence with positive margins [29]. Therefore, PN could have contributed to the higher RFS compared to RFA. Although PN showed higher RFS, RFA was also found to be highly effective, as the 5-year RFS was more than 85% in patients regardless of CKD stage. Furthermore, the recurrences after RFA (*n* = 11) were well controlled by re-RFA (*n* = 2), cryotherapy (*n* = 3), RN (*n* = 1), and sunitinib (*n* = 3). This was also confirmed by a remarkably low cancer-specific mortality. Therefore, if surgery is not considered the best option in consideration of the features of cancer or a patient’s conditions, RFA should be considered a good alternative.

In the present study, before performing the propensity score matching, the 5-year CSS rates were 99.8% and 97.0% in the PN and RFA groups, respectively. After performing the propensity score matching, the 5-year CSS rates were 99.6% and 100% in the PN and RFA groups, respectively. Various studies have reported a low mortality rate in patients with small renal tumors. Andrews et al. reported that the 5-year CSS rates for patients with a cT1a renal mass treated with PN, RFA, and cryoablation were 99.3%, 95.6%, and 100%, respectively [4]. These results are consistent with our data.

Regarding the OS rate, PN was associated with a significantly higher five-year OS rate compared to RFA in all patients with tissue-proven malignancies. Similar findings were reported by other studies. For example, Andrews et al. reported that the 5-year OS rates for T1a patients treated with PN and RFA were 92% and 72%, respectively. Patients treated with RFA were significantly more likely to die from any cause compared with those treated with PN [4]. In this study, we included patients with CKD who were more likely to have other comorbidities or poorer conditions. Additionally, it is highly likely that patients with poor conditions, including older age and reduced eGFR, underwent RFA. Therefore, such patients with poor conditions or other comorbidities may have undergone RFA, and this selection bias may have led to a higher OS rate in the PN group. After propensity score matching with variables including age and pre-treatment eGFR, there was no significant difference in the OS rate between the RFA and PN groups in patients with tissue-proven malignancy regardless of CKD stage. This finding suggests that adjusting the selection bias through propensity score matching led to no difference in OS between the two groups.

This study has a few limitations. First, this was a retrospective, non-randomized study. The likelihood of a selection bias between the PN and RFA groups cannot be completely excluded. In addition, tumor complexity using a nephrometry score was not described, and tumor locations were not matched between the PN and RFA groups in this study. Additionally, despite propensity score matching having been performed using eGFR and tumor size, there was a significant difference in eGFR between the two groups in the propensity score matched cohorts with CKD stage 3 or higher. It may be due to a big difference in renal function between the two groups and the small number of patients with CKD stage 3 or higher that could not be calibrated after matching. Therefore, we could not evaluate functional outcomes in patients with CKD stage 3 or higher using propensity score matching. Instead, we compared serial eGFR and the mean percentage change in eGFR at 1, 2, and 3 years post-treatment between the two groups in patients with CKD stage 3 or higher. Second, adverse oncological outcomes, especially cancer-specific mortality, were relatively rare in this cohort. This may have weakened the statistical power of our analyses. Third, we could not consider major comorbidities such as the Charlson Comorbidity Index. Furthermore, we could not confirm the specific pathological results such as pathological T staging and Fuhrman grade in patients treated with RFA. Although these factors could affect survival, we could not consider these factors in the survival analysis. Fourth, the follow-up period was not sufficient to estimate long-term oncological outcomes. The remarkably low cancer-specific mortality might be due to the insufficient follow-up period. Despite these limitations, this study provides the first comparison of oncological and functional outcomes after PN and RFA in patients according to the CKD stage.

## 5. Conclusions

In conclusion, PN is superior to RFA in terms of RFS rate in CKD patients with small renal tumors. However, patients with small renal tumors rarely die of renal cancer, and other comorbidities play important roles in the treatment outcomes in CKD patients. As there was no significant difference in the deterioration of renal function between the two treatments, RFA could be an alternative treatment modality in patients who are poor candidates for surgery owing to CKD.

## Figures and Tables

**Figure 1 diagnostics-12-02292-f001:**
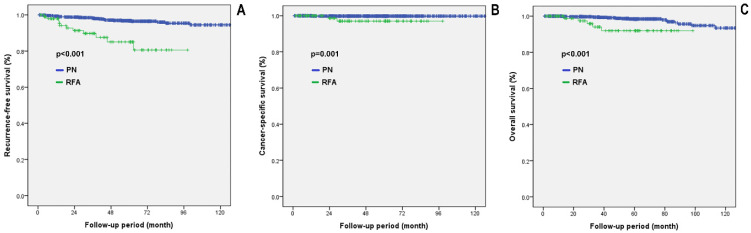
Survival analysis in all patients with cT1a tissue-proven malignancy. (**A**) RFS, recurrence-free survival; (**B**) CSS, cancer-specific survival; (**C**) OS, overall survival. PN: partial nephrectomy; RFA: radiofrequency ablation.

**Figure 2 diagnostics-12-02292-f002:**
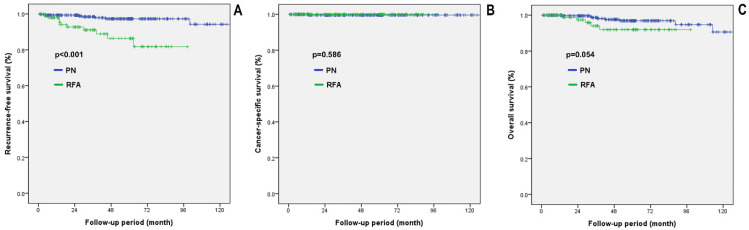
Survival analysis in propensity score matched cohorts. (**A**) RFS, Recurrence-free survival; (**B**) CSS, Cancer-specific survival; (**C**) OS, Overall survival; PN, Partial nephrectomy; RFA, Radiofrequency ablation.

**Figure 3 diagnostics-12-02292-f003:**
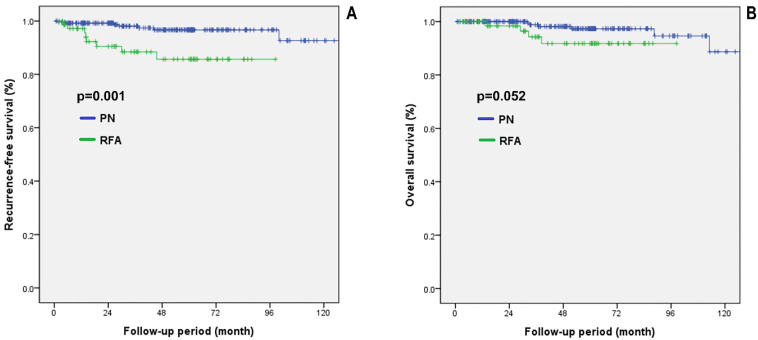
Survival analysis in propensity score matched cohorts with CKD stage 2. (**A**) RFS, recurrence-free survival; (**B**) OS, overall survival. PN: partial nephrectomy; RFA: radiofrequency ablation.

**Figure 4 diagnostics-12-02292-f004:**
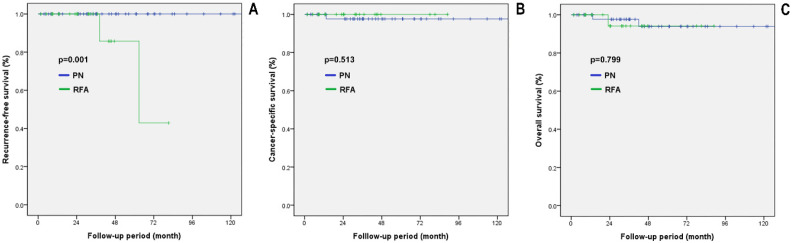
Survival analysis in propensity score matched cohorts with CKD stage 3 or higher. (**A**) RFS, recurrence-free survival; (**B**) CSS, cancer-specific survival; (**C**) OS, overall survival. PN: partial nephrectomy; RFA: radiofrequency ablation.

**Table 1 diagnostics-12-02292-t001:** Patients’ demographics and tumor characteristics.

Variables	CKD Stage 2 (60 ≤ eGFR < 90)			CKD Stage 3 or Higher (eGFR < 60)		
	PN (*n* = 1060)	RFA (*n* = 135)	*p*	PN (*n* = 84)	RFA (*n* = 53)	*p*
Age, mean ± SD	56.0 ± 11.1	63.0 ± 12.4	**<0.001**	64.7 ± 9.7	66.5 ± 11.7	0.348
Sex, male, *n* (%)	803 (75.8)	100 (74.1)	0.669	62 (73.8)	41 (77.4)	0.640
HTN, *n* (%)	416 (39.2)	62 (45.9)	0.223	57 (67.9)	43 (81.1)	0.088
DM, *n* (%)	150 (14.2)	22 (16.3)	0.607	30 (35.7)	18 (34.0)	0.817
Pre-treatment eGFR	78.6 ± 7.6	75.9 ± 7.9	**<0.001**	51.3 ± 9.3	33.2 ± 18.8	**<0.001**
Clinical tumor size (cm)	2.37 ± 0.80	2.13 ± 0.83	**0.001**	2.52 ± 0.75	2.20 ± 0.85	**0.022**
Tumor histology, *n* (%)			**<0.001**			**<0.001**
Unknown (not biopsied)	0 (0)	47 (34.8)		0 (0)	19 (35.8)	
Benign	35 (3.3)	12 (8.9)		2 (2.4)	9 (17.0)	
Clear cell RCC	845 (79.7)	62 (45.9)		69 (82.1)	22 (41.5)	
Non-clear cell RCC	177 (16.7)	14 (10.4)		13 (15.5)	3 (5.7)	
Other type malignancy	3 (0.3)	0 (0)		0 (0)	0 (0)	
Follow-up period (in months), mean ± SD	52.4 ± 33.1	51.0 ± 36.0	0.649	52.4 ± 35.7	43.3 ± 39.0	0.166

CKD: chronic kidney disease; eGFR: estimated glomerular filtration rate; PN: partial nephrectomy; RFA: radiofrequency ablation; HTN: hypertension; DM: diabetes mellitus; Cr: creatinine; RCC: renal cell carcinoma; SD: standard deviation.

**Table 2 diagnostics-12-02292-t002:** Patient demographics and tumor characteristics in propensity score matched cohorts.

Variables	CKD Stage 2 (60 ≤ eGFR < 90)			CKD stage 3 or Higher (eGFR < 60)		
	PN (*n* = 270)	RFA (*n* = 135)	*p*	PN (*n* = 53)	RFA (*n* = 53)	*p*
Age, mean ± SD	62.8 ± 10.9	63.0 ± 12.4	0.861	66.6 ± 8.8	66.5 ± 11.7	0.940
Sex, male, *n* (%)	195 (72.2)	100 (74.1)	0.693	38 (71.7)	41 (77.4)	0.504
HTN, *n* (%)	124 (45.9)	62 (45.9)	0.795	38 (71.7)	43 (81.1)	0.237
DM, *n* (%)	50 (18.5)	22 (16.3)	0.495	20 (37.7)	18 (34.0)	0.684
Pre-treatment eGFR	76.3 ± 8.2	75.9 ± 7.9	0.664	48.6 ± 10.6	33.2 ± 18.8	**<0.001**
Clinical tumor size (cm)	2.19 ± 0.74	2.13 ± 0.83	0.513	2.27 ± 0.70	2.20 ± 0.85	0.668
Tumor histology, *n* (%)			**0.001**			**0.002**
Unknown (not biopsied)	0 (0)	47 (34.8)		0 (0)	19 (35.8)	
Benign	8 (3.0)	12 (8.9)		2 (3.8)	9 (17.0)	
Clear cell RCC	213 (78.9)	62 (45.9)		46 (86.8)	22 (41.5)	
Non-clear cell RCC	49 (18.1)	14 (10.4)		5 (9.4)	3 (5.7)	
Follow-up period (in months), mean ± SD	51.4 ± 34.2	51.0 ± 36.0	0.914	54.3 ± 40.3	43.3 ± 39.0	0.158

CKD: chronic kidney disease; eGFR: estimated glomerular filtration rate; PN: partial nephrectomy; RFA: radiofrequency ablation; HTN: hypertension; DM: diabetes mellitus; Cr: creatinine; RCC: renal cell carcinoma; SD: standard deviation.

**Table 3 diagnostics-12-02292-t003:** Pre- and post-treatment estimated renal function according to treatment method in patients with CKD stage 2.

Variables	CKD Stage 2 (60 ≤ eGFR < 90) All Patients			CKD stage 2 (60 ≤ eGFR < 90) Propensity Score Matched Cohorts		
	PN (*n* = 1060)	RFA (*n* = 135)	*p*	PN (*n* = 270)	RFA (*n* = 135)	*p*
eGFR Pre-treatment	78.6 ± 7.6	75.9 ± 7.9	**<0.001**	76.3 ± 8.2	75.9 ± 7.9	0.668
eGFR at 1 year post-treatment	58.8 ± 32.9	49.6 ± 34.8	**0.002**	56.4 ± 31.7	49.6 ± 34.8	0.057
% Change eGFR at 1 year post-treatment	−4.3 ± 14.4	−5.5 ± 15.7	0.458	−6.1 ± 14.1	−5.5 ± 15.7	0.713
CKD upstaging at 1 year post-treatment, *n*, (%)	85 (8.0)	19 (14.1)	**0.019**	35 (13.0)	19 (14.1)	0.756
Upstaging to stage 3	84 (7.9)	19 (14.1)		35 (13.0)	19 (14.1)	
Upstaging to stage 4	1 (0.1)	0 (0.0)		0 (0)	0 (0.0)	
Upstaging to stage 5	0 (0)	0 (0.0)		0 (0)	0 (0.0)	

eGFR: estimated glomerular filtration rate; CKD: chronic kidney disease; PN: partial nephrectomy; RFA: radiofrequency ablation.

**Table 4 diagnostics-12-02292-t004:** Pre- and post-treatment estimated renal function according to treatment method in patients with CKD stage 3 or higher.

Variables	CKD Stage 3 or Higher (eGFR < 60)		
	PN (*n* = 84)	RFA (*n* = 53)	*p*
eGFR Pre-treatment	51.3 ± 9.3	33.2 ± 18.8	**<0.001**
eGFR at 1 year post-treatment	38.3 ± 23.6	23.1 ± 23.1	**<0.001**
% Change eGFR at 1 year post-treatment	−5.4 ± 17.2	−12.2 ± 23.4	0.132
eGFR at 2 years post-treatment	40.2 ± 22.6	25.0 ± 24.9	**<0.001**
% Change eGFR at 2 years post-treatment	−8.0 ± 24.3	−16.8 ± 27.6	0.093
eGFR at 3 years post-treatment	48.5 ± 16.8	32.5 ± 22.5	**<0.001**
% Change eGFR at 3 years post-treatment	−7.3 ± 27.9	−22.0 ± 28.4	**0.012**
CKD upstaging at 3 years post-treatment, *n*, (%)	6 (7.1)	9 (17.0)	0.072
Upstaging to stage 4	2 (2.4)	3 (5.7)	
Upstaging to stage 5	4 (4.7)	6 (11.3)	

eGFR: estimated glomerular filtration rate; CKD: chronic kidney disease; PN: partial nephrectomy; RFA: radiofrequency ablation.

## Data Availability

The dataset used and/or analyzed during the current study is available from the corresponding author upon reasonable request.

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
