# Peer review of "A Comparison of Functional and Oncologic Outcomes between Partial Nephrectomy and Radiofrequency Ablation in Patients with Chronic Kidney Disease after Propensity Score Matching"

_diagnostics, 2022, doi:10.3390/diagnostics12102292_

Round 1

Reviewer 1 Report

The authors shoud be congratulated for the work, however I have several concerns:

o   The rationale of the study is very interesting expecially with the aim to preserve nephrons ( kidney sparing procedure ) supporting the idea that each nephron counts ( the authors could cite this article 10.7717/peerj.7640 whose main aim is to assess that the reduction in nephron count might influence the global renal status. Nevertheless, despite the originality of the topic a direct comparison between these techniques is difficult to be understood; in fact, indications and approach are different for these surgeries according to the latest EAU guidelines because they depend on both patient’s and tumour’s characteristics (Ie. Cryotherapy, radiofrequency are reserved to fragile patients and are not indicated for hilar located lesione etc ). Please explain this concept in the introduction and methods.

o   Can you add any information about the median ischemia time during partial nefrectomy? Expecially with small renal masses, an “off-clamp” technique can be offered to maximize the preservation of renal function. [DOI: 10.1007/s00345-019-02879-4]. Infact, garanting the most intraoperative tissue perfusion is the primary factor related to postoperative functional outcomes, as also demonstrated during kidney transplantation. [DOI:10.23736/S0393-2249.18.03278-2]. You could mention these topics.

o   Considering that patients with CKD are at increased risk for progressive decline in renal function after active treatment, in some selected patients active surveillance might be the best nephron sparing approach, reserving treatment only when triggers for delayed intervention occurr. [DOI: 10.23736/S0393-2249.20.03870-9].

o   What was, in centimeters, the dimensional cut off you used to treat SRM with RFA? What was the percentage of patients that needed and additional ablation after the first treatment?

Reviewer 2 Report

The authors compared partial nephrectomy to radiofrequency ablation in the treatment of small renal tumors in patients with different stages of chronic kidney disease. The study is well done with regards to methodology and references are appropriately cited. The statistical measures are superb and clearly reflect the findings. Can the authors elaborate on a subgroup of dialysis patients (eGFR < 15-20) who underwent these procedures and give us the results of these two procedures on the primary and secondary outcomes? 

Reviewer 3 Report

I congratulate the Authors on their interesting analysis of their genuine cohort of cT1a tumours treated with a kidney-sparing/ focal approach.

Please find my comments below:

I would supplement the abstract with factors included in propensity score matching.

The introduction provides in-line insight into the rationale of the study and the idea behind it.

In general, the methodology design is decent and consecutive. What raises my concerns is:

-          Why the only comorbidities you tried to adjust for were HT and DM? Major comorbidities that might substantially confound survival like coronary disease are lacking evaluation. why you have not used Charlson Comorbidity Index?

-          Despite significant differences in pathological characteristics, this variable was not included in the matching process. Why?

Also, how was recurrence defined? Was every radiological suspicion of recurrence verified pathologically?

Results

Table 1 and Table 2 ‘CKD stage 3 or high’ I guess it should be ‘or higher ‘

As I understand Table 4 depicts the same analysis presented in Table 3 but for stage 3 CKD and higher. Table 3 however presents both PSM-matched cohorts and entire cohorts. Why this has not been done for Table 4 (stage 3 and higher?)

After PSM-matching you found no significant difference in OS  although p=0.054 shows a clear tendency towards statistical significance. I understand that a test failing conservative statistical significance is insignificant regardless of the exact p-value, but still, I would keep myself very cautious when delivering categorical statements like no impact of RFA on overall survival at all. After all, a borderline p-value might also implicate insufficient adjustment for comorbidities which I mentioned above. The matter of RFS not translating onto CSS might simply come from the follow-up limitation.

Figures are low quality and extremely hard to read. Please replace them with improved-resolution figures.

Discussion

The discussion is generally well-written, clear and consecutive. I find it interesting. I would consider supplementing the limitations section with the issues mentioned above.

Also, let the first sentences of the discussion deliver the main outcomes of the study. Delivering the rationale of the analysis (which they do now) should rather be included in the introduction only.

Round 2

Reviewer 1 Report

Thanks for your answers. 

Reviewer 3 Report

The Authors have answered all my questions and followed all my suggestions, I believe the paper can be proceeded towards acceptance now.